# Enhancing the Microstructure and Sustainability of Ultra-High-Performance Concrete Using Ultrafine Calcium Carbonate and High-Volume Fly Ash under Different Curing Regimes

**Norzaireen Azmee** [1], **Yassir M. Abbas** [2], **Nasir Shafiq** [1], **Galal Fares** [2], **Montasir Osman** [1] and **M. Iqbal Khan** [2,*]

[1] Department of Civil and Environmental Engineering, Universiti Teknologi PETRONAS, Perak 32610, Malaysia; norzaireen@gmail.com (N.A.); nasirshafiq@utp.edu.my (N.S.); montasir.ahmedali@utp.edu.my (M.O.)
[2] Department of Civil Engineering, King Saud University, Riyadh 800-11421, Saudi Arabia; yabbas@ksu.edu.sa (Y.M.A.); galfares@ksu.edu.sa (G.F.)
* Correspondence: miqbal@ksu.edu.sa; Tel.: +966-1-46-76-920

**Abstract:** In current practice, the performance-based concrete mix (PBCM) approach has become quite popular because it enhances the quality of materials that are fundamentally necessary for a particular situation. In the present study, experimental analysis is performed to determine the optimal mechanical properties and microstructural characteristics of concrete for sustainable development and cost effectiveness. Specifically, a mixture of high-volume fly ash (FA) and ultrafine calcium carbonate (UFCC) is investigated as a partial substitution of cement. For optimizing the concrete's performance, various curing regimes are applied to evaluate the best conditions for obtaining ideal mechanical and microstructural properties. The results show that concrete containing 10% UFCC with a mean particle size of 3.5 μm blended with 40% FA yielded the best performance, with an enhancement of 25% in the compressive strength in the early age. Moreover, the UFCC improved the compactness and refined the interstitial transition zone (ITZ). However, the effects of the different curing methods on the concrete's strength were insignificant after 28 days.

**Keywords:** high-volume fly ash; ultrafine calcium carbonate; heat curing effects; microstructural characteristics; interfacial transition zone



## 1. Introduction

After water, concrete is the second most consumed material, with about 4.2 billion tons produced worldwide in 2019 [1,2]. Over the last 20 years, tremendous effort has been made in advancing concrete technology, with remarkable milestones achieved. Researchers worldwide have attempted to develop high-strength concrete materials with excellent performance. In particular, recent advancements have significantly improved the mechanical properties and durability characteristics of concrete. As a result, high-performance concrete (HPC) and ultra-high-performance concrete (UHPC) are currently used in many infrastructure projects [3]. The adaptability of such technological advancements in industrial practices has been explored; however, the high construction costs, limited codes, optimization challenges, and unconventional production techniques, together with limited obtainable resources, have hindered the practical application of these products in modern building technology, particularly in emerging economies [4]. UHPC mixes contain a high cement content, from about three to five times that generally used in conventional concrete mixes [5,6]. In addition, the low water-to-binder ratio (W/B) used in UHPC mixes causes a large part of the cement to remains unhydrated, which is considered to be a waste of resources [7–9]. As reported in many studies, cement manufacturing is responsible for

approximately 8% of global $CO_2$ emissions, which have severe impacts on global warming [10–13]. For addressing the issues raised and various technological gaps, abundant research has been conducted to reduce the cement content in UHPC without compromising its mechanical properties. The use of highly active supplementary cementitious materials (SCMs) in the UHPC mix design would decrease the amount of cement needed, which can lower the cost and increase the eco-efficiency of UHPC mixes.

Excellent durability is achieved using UHPC mixes containing various types of SCMs. Among them, silica fume (SF) is considered the most favorable pozzolanic material for developing UHPC because it can be used as a partial substitute in cement at a weight ratio of 25–30% [14,15]. In addition, SF has the ability to fill pores in the cement paste, which enhances the rheological characteristics and the formation of hydration products, resulting in high-performance concrete (HPC) [16,17]. However, the substantial cost, scarcity, and maldistribution of SF limit its use, particularly in the context of sustainability. On the contrary, fly ash (FA), a byproduct of coal burning, is available in enormous amounts worldwide at low cost. This material is considered to be one of the best options for improving the dispersibility of SF and a portion of ordinary Portland cement (OPC) in UHPC. Thus, FA is commonly used as a substitute for 20–50% of the cement in conventional and high-strength concretes [3,10,18,19].

Generally, for producing low-cement concrete, a high FA replacement volume is needed. FA causes a reduction in the early age concrete strength compared with that in a control mix composed of 100% cement [20]. Recent studies observed that FA contributes to enhanced concrete strength when the W/B is kept low [21,22]. An earlier study conducted by Lam et al. [23] demonstrated that a concrete mix with 45% FA at a W/B of 0.5 resulted in about a 30% reduction in the 28-day compressive strength. Further decreases in the W/B lowered the reduction level to 17%. Based on these findings, it is anticipated that high-volume FA (HVFA) in performance-based concrete mixes will contribute to better strength performance.

Generally, ultrafine powders accelerate the hydration process of cement, which is governed by the size and shape of refined particles. Hence, a dense microstructure is formed [24]. Su et al. [25] reported that UHPC containing steel fibers and different nanomaterials including nano-$CaCO_3$, nano-$SiO_2$, nano-$TiO_2$, and nano-$Al_2O_3$ showed excellent ductility and blast resistance. Soliman et al. [26] prepared a green UHPC with a compressive strength of 200 MPa by incorporating nano-glass powder in the mix, which provided technological, economic, and environmental advantages over conventional UHPC. Burroughs et al. [27] reported that limestone at minor replacement levels acts as an effective inert filler and does not negatively affect the UHPC properties. Another method used to increase the early strength performance and the quality of UHPC is subjecting it to high-temperature curing [15,28]. For enhancing the mechanical properties, heat treatment at 60 °C and 90 °C is applied to UHPC for 48–72 h after setting. Graybeal and Hartmann [29] investigated the effects of different curing methods on the compressive strength of UHPC and found significant variations in compressive strength. The thermal curing technique was found to be the most suitable for precast operations. However, it incurs high energy consumption and cost. Hence, its acceptance for field application is impractical.

Recently, Zbigniew [30] published a comprehensive review of the effects of fly ash and slag combined with lime on concrete properties. The study focused on preparing blended cement by introducing high-level functional additives, such as nano-materials, to concrete composition, and studying their chemical and mechanical activation. It is also recognized that many researchers have made efforts to synergize the effects of fly ash with limestone on the durability of concrete [30]. Additionally, Nowoświat and Gołaszewski [31] have recently reported their study on the impacts of calcareous fly ash (CFA) obtained from various sources on mortar rheological properties. The research findings demonstrated the adverse effects of CFA on mortar workability and discouraged supporting its use in concrete manufacturing. In addition, Huailiang et al. [32] investigated the stress-strain behavior of fly ash and slag alkali-activated concrete tested under uniaxial compression.

Spontaneous failure of specimens was observed once they reached peak stress, which was more pronounced with the high-grade concrete mixes.

The development of cost-effective and environmentally friendly HPC is crucial for achieving a wider acceptance by the construction industry. Hence, the main objective of the present study is to develop a concrete mix design that satisfies these requirements. In the mix-design approach employed, a significant portion of SCM, up to 50%, was used. The blended material was composed of ultrafine calcium carbonate (UFCC) with a mean particle size (D50) less than 3.5 μm and a high FA volume, with a D50 of about 24 μm. Such a type of SCM can be regarded as a cost-effective and environmentally friendly material. For enhancing the mechanical properties and microstructural characteristics, various curing conditions were employed such as heat curing, ambient curing, and water curing.

## 2. Materials and Methods

### 2.1. Material Properties

In developing the concrete mix proportions, a low cement content of 50% was kept in preparing the binding material using SCM. The SCM content, at 50%, included FA only, FA plus UFCC, and FA plus SF. These fine materials were obtained by local suppliers in Malaysia. In the current research, the X-ray fluorescence method (XRF) was employed to investigate the chemical composition of the OPC, FA, SF, and UFCC. The XRF analysis results are given in Table 1. The particle size analysis was studied using scanning electron microscopy (SEM); the obtained images are shown in Figure 1. Moreover, crushed granite with a maximum grain size of 10 mm acquired from a local quarry in Perak, Malaysia, was utilized as a coarse aggregate (Figure 2a). Additionally, fine river sand with a fineness modulus of 2.8 obtained from a local deposit near Ipoh, Malaysia, was used as fine aggregate (Figure 2a). A low W/B ratio of 0.16 was used in the concrete mix. Therefore, a polycarboxylic ether-based superplasticizer (SP) was applied to achieve the desired workability. A straight steel fiber content of 1.0% by volume was investigated in this study. The physicomechanical properties of the steel fibers are shown in Table 3. Additionally, Figure 2b depicts the microscopic size measurement of the micro steel fiber.

**Table 1.** Physicochemical properties of OPC, FA, SF, and UFCC.

|  |  | OPC | FA | SF | UFCC |
|---|---|---|---|---|---|
|  | $SiO_2$ | 12.38 | 36.41 | 90.4 | 0.19 |
|  | $Al_2O_3$ | 2.86 | 16.95 | 0.71 | 0.07 |
|  | $Fe_2O_3$ | 5.32 | 20.54 | 1.31 | 0.05 |
|  | $TiO_2$ | 0.17 | 1.59 | - | - |
|  | MnO | 0.09 | 0.18 | - | - |
| Chemical | MgO | 0.99 | 2.26 | - | 0.8 |
| composition (%) | CaO | 73.50 | 14.4 | 0.45 | 54.6 |
|  | $Na_2O$ | - | 1.03 | - | - |
|  | $K_2O$ | 0.80 | 2.23 | - | - |
|  | $P_2O_5$ | 0.44 | 1.35 | - | - |
|  | $SO_3$ | 3.11 | 2.19 | 0.41 | - |
|  | L.O.I. | 4.7 | 1.7 | 5.4 | 43 |
| Physical properties | D50 (μm) | 20.8 | 24.4 | 0.16 | 3.5 |

D50 = mean particle size.



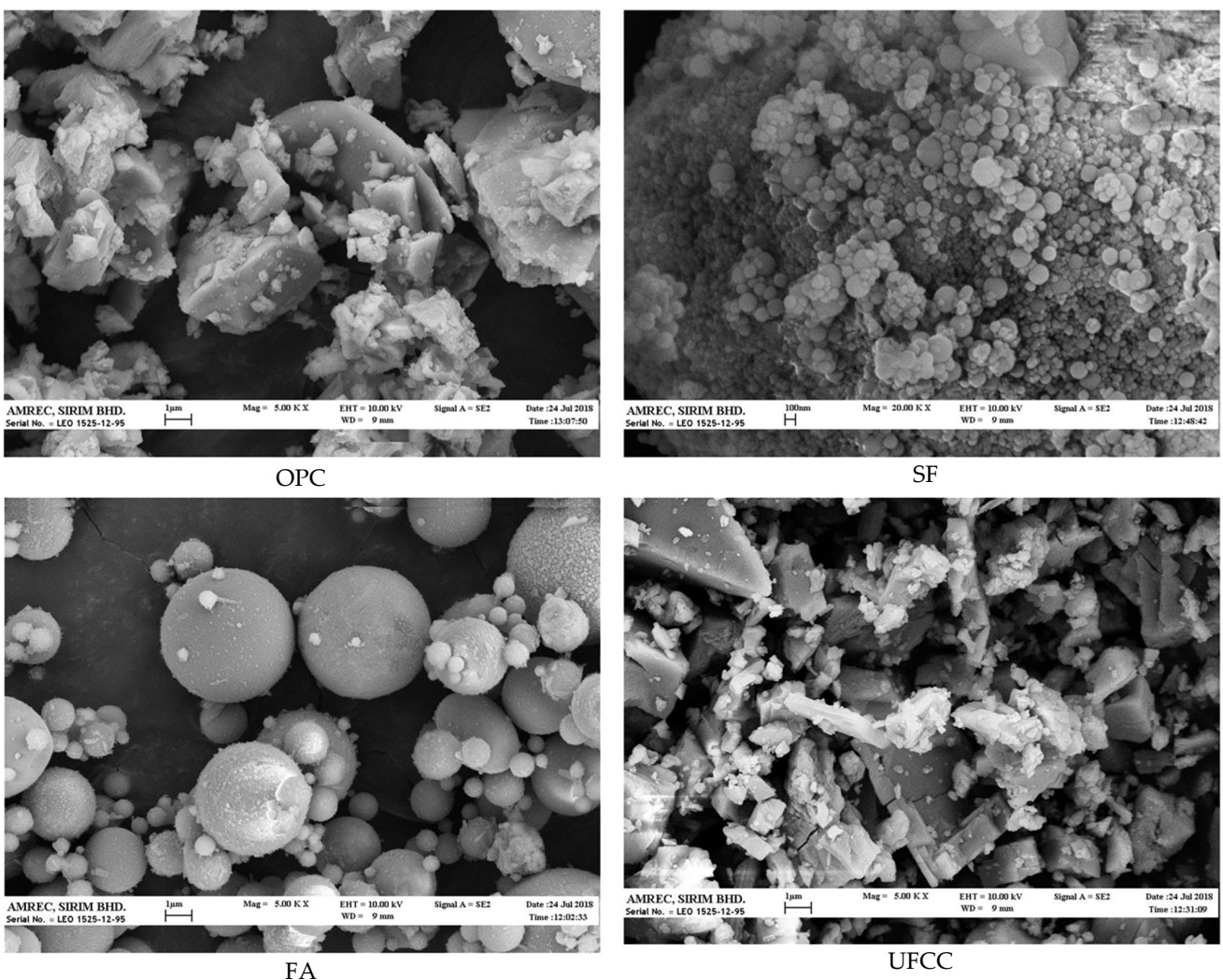

**Figure 1.** SEM images of OPC, SF, FA, and UFCC.

### 2.2. Concrete Mix Proportions, Specimen Casting, and Curing

A trial mix design method was adopted for this program; the details are given in Table 2. For all concrete mixes, the W/B was kept constant at 0.16. Therefore, a dosage of SP between 1.5 wt.% and 4 wt.% binder mass was added to achieve the target slump value (i.e., 80 mm ± 10 mm).

**Table 2.** Concrete mix compositions.

| Mix ID | Binder | % Binder | | | | CA | Sand | % Fiber |
|---|---|---|---|---|---|---|---|---|
| | | OPC | FA | SF | UFCC | | | |
| M0 | 900 | 100 | - | - | - | 930 | 620 | 1 |
| M1 | 900 | 50 | 50 | - | - | 930 | 620 | 1 |
| M2 | 900 | 50 | 40 | 10 | - | 930 | 620 | 1 |
| M3 | 900 | 50 | 40 | - | 10 | 930 | 620 | 1 |

All quantities are in kg/m$^3$ unless otherwise stated.

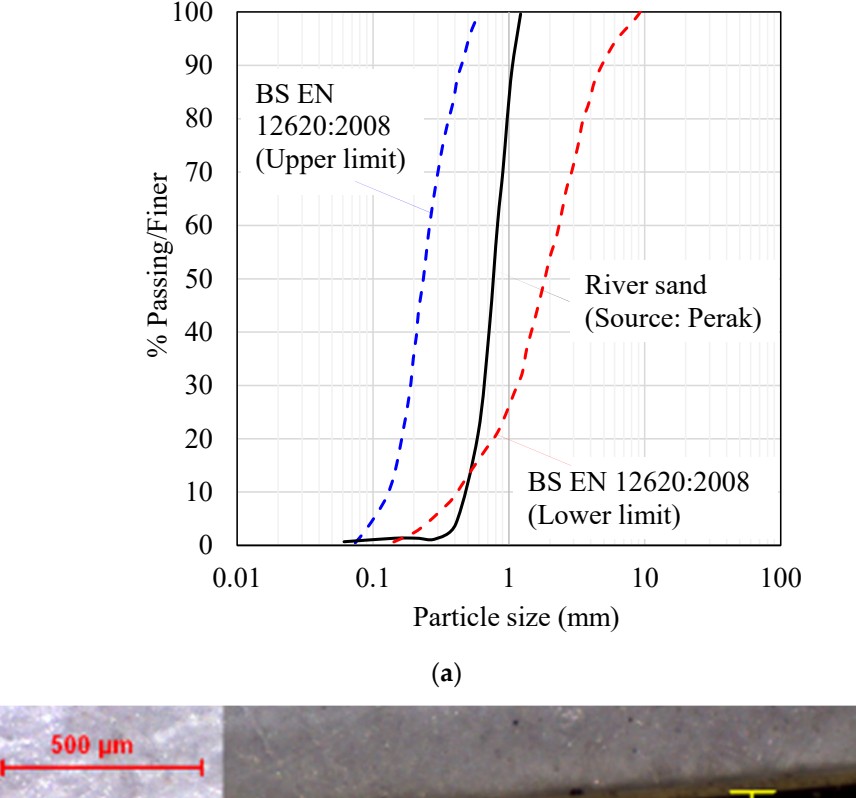

(**a**)

(**b**)

**Figure 2.** (**a**) Particle size distribution of the sand; (**b**) microscopic verification of micro steel fiber.

**Table 3.** Characteristics of the steel fiber.

| Fiber Properties | Straight End |
|---|---|
| Length, mm | 20 |
| Diameter, mm | 0.2 |
| Aspect ratio ($l/d$) | 100 |
| Tensile strength, MPa | >2300 |

For mixing and batching of the concrete mixes, a 60 L capacity concrete mixer was used. The batching and mixing operation was performed according to the procedure described in Table 4. The workability test results for all mixes fulfilled the Class Ca requirement, as suggested in NF P18-470 [33]. All samples were cast in steel molds; after casting, the samples were carefully wrapped in a plastic membrane to prevent drying. All samples were demolded after 24 h of casting and subjected to three different curing regimes, as listed in Table 5.

**Table 4.** Concrete mixing times.

| Mixing Procedure | Time (min) |
|---|---|
| Add aggregates + 3% moisture content | 3 |
| Add binder | 5 |
| Add 50% water + 50% SP | 5 |
| Add 50% water + 50% SP | 5 |
| Add steel fiber | 2 |

**Table 5.** Different curing regimes.

| | Curing Method | Testing (Days) |
|---|---|---|
| WC | Water curing at room temperature until the date of testing | |
| AC | Ambient curing until the date of testing | 1, 7, 14, 28, 90 |
| HC | Heat curing at 90 °C for 8 h in a water bath, and specimens were formerly relocated in water curing tank at room temperature until the date of testing | |

### 2.3. Preparation for Compressive Strength and Microstructure Analysis

All specimens were subjected to compressive strength testing on a 100 mm cube, conforming to BS EN 12390-3 [34]. All results reported in this study represent the average of three tested samples at curing ages of 1, 7, 14, 28, and 90 days. For a clear understanding of the effects of the curing conditions on the strength development of different concrete mixes, thermogravimetry analysis (TGA) was performed on the paste samples to evaluate the calcium hydroxide (CH) content. For performing TGA, a small fraction from the hardened paste sample was taken. The broken pieces of the paste were soaked in acetone for seven days to stop the hydration process. Afterward, the specimens were oven-dried at 60 °C for 8 h, and were then ground by hand using an agate mortar and pestle to enable passage through a 150 μm sieve. SEM supported by energy-dispersive X-ray (EDX) analysis was used to evaluate the effects of the different SCM on the microstructure properties of HPC.

### 2.4. Preparation of Paste Samples for Multispectral SEM Analysis

In this part of the analysis, thin sections were obtained from the paste sample after a curing time of 28 days. For this preparation, a cutting and grinding machine with a diamond blade (Geoform 102, Metkon Instruments, Inc., Bursa, Turkey) was used to obtain a thickness of not more than 1 mm, as illustrated in Figure 3.

### 2.5. Direct Tensile and Flexural Strengths Testing Methods

In contrast to that for compression, no standard is available for direct tensile testing of concrete. Several researchers have suggested various test setups with different specimen shapes and gripping systems. In this study, the direct tensile test was conducted using dog-bone shaped samples recommended by JSCE [35]. The specimens were tested under an increasing deformation until failure by using a universal testing machine (GOTECH Testing Machines, Inc., Taichung City, Taiwan) with a maximum load capacity of 200 kN. The loading was applied under displacement control at a rate of 0.01 mm/s. Figure 4 shows the test setup and the instrumentation used. Three samples were replicated in each test. The tensile limit and post-peak tensile behavior were determined by using the average curve from the test results.

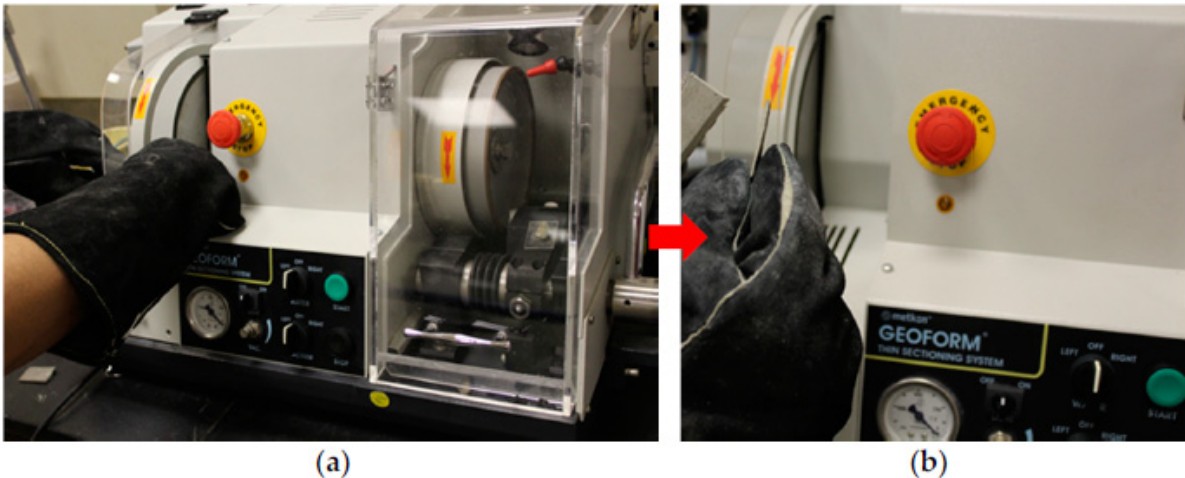

**Figure 3.** Preparation of cement paste samples for multispectral analysis. (**a**) Cutting and preparing the thin sections; (**b**) the obtained thin sections.

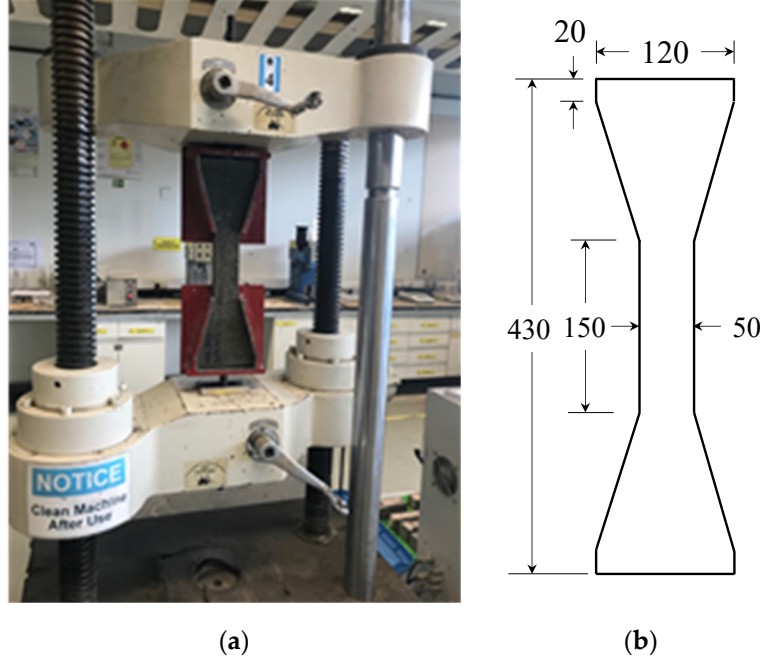

(**a**)                                    (**b**)

**Figure 4.** (**a**) Direct tensile test setup and (**b**) dog-bone specimen with thickness of 30 mm (dimensions are in millimeters).

Moreover, standard-sized prisms of 100 mm × 100 mm × 500 mm were used and were tested after 28 days of curing. Three specimens were tested for each mix, and an average of the three tests was used to obtain the results. A four-point flexural testing machine with two bottom supportive rollers and two upper ones supported on an articulated cross-member was used in this test. Because this research focuses on HPC, the testing procedure followed the standard recommended in NF P18-470 [33]. During the test, each specimen was loaded under an actuator displacement control of 0.0042 mm/s.

## 3. Results and Discussion

### 3.1. Investigation of the Effects of SCM and Curing Method on the Compressive Strength

For all concrete mixes, compressive strength results were obtained within the range of 80–162 MPa when subjected to different curing conditions. Figure 5 shows that different

curing methods had considerable effects on concrete compressive strength. All heat-cured (HC) specimens achieved very high compressive strength of more than 100 MPa at the seven-day point. Increments in the compressive strength of about 5% were observed after seven days. At 90 days of HC, all blended concrete (mixes M1, M2, and M3) showed a slight reduction (1.3–4.8%) in compressive strength. Such trends were consistent in all concrete mixes using different SCM combinations. A possible reason for achieving high compressive strength at seven days of HC is the enhanced and accelerated pozzolanic reactivity. The pozzolanicity of materials could be accelerated by the high temperature, as energy input and moisture during HC. The enhanced pozzolanicity resulted in a refined microstructure of the hydrated product. Therefore, a faster strength development process could be clarified [36].

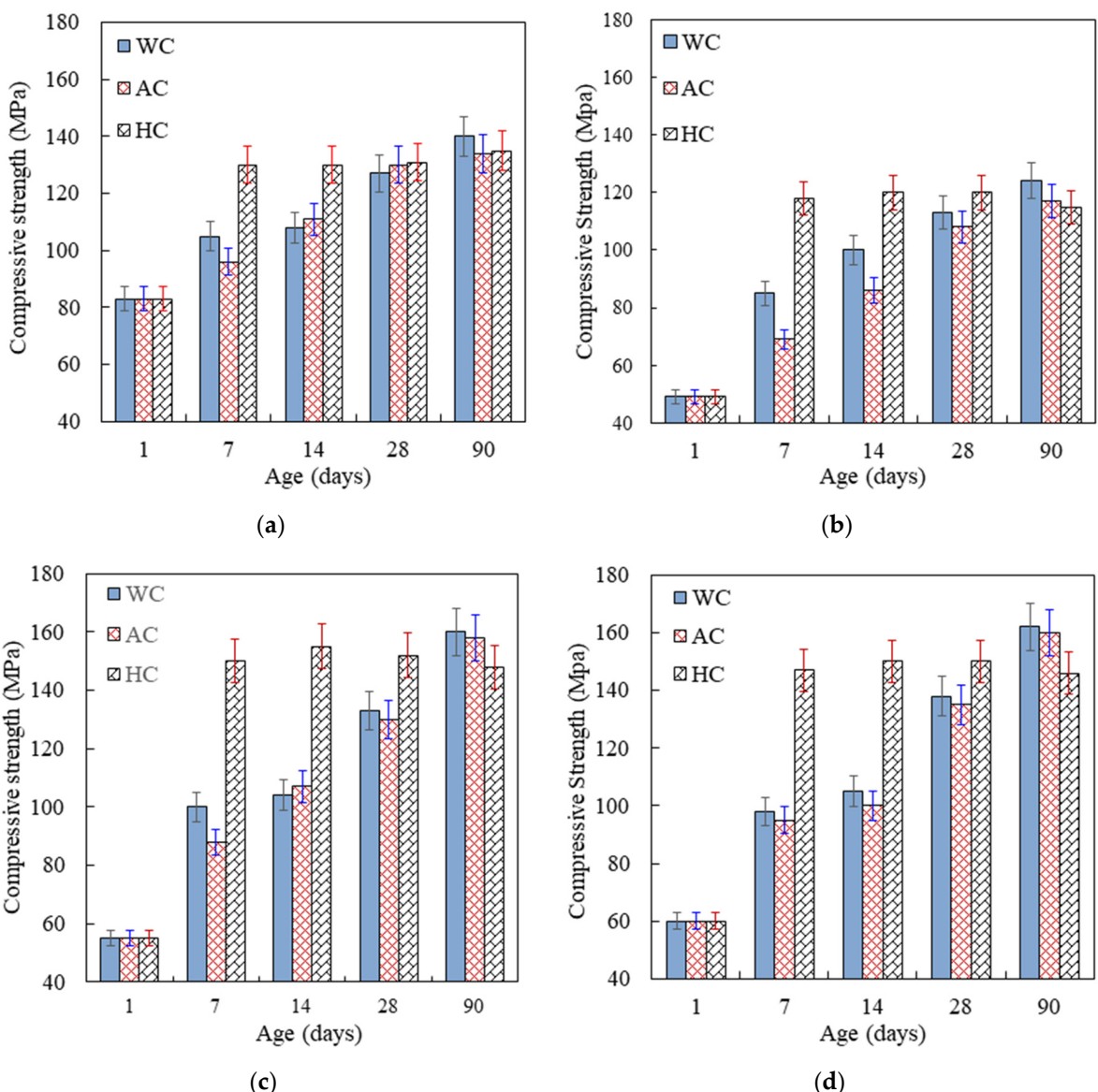

**Figure 5.** Compressive strength of ultra-high-performance concrete (UHPC) at different curing regimes: (**a**) M0, (**b**) M1, (**c**) M2, and (**d**) M3.

The XRD results of pastes cured for 28 days then heat treated under different temperatures of 20 °C (HT20), 60 °C (HT60), and 90 °C (HT90) (Figure 6) showed reduced peak intensities of both $C_2S$ and $C_3S$ tested at 28 days compared with the sample cured under normal conditions. Pastes treated under HC at 90 °C (HT90) displayed the lowest $C_2S$ and

C$_3$S values, indicating a higher degree of cement hydration at higher curing temperature. HT treatment promotes significant cement hydration and pozzolanic reactions and improves the microstructural refinement [36]. Moreover, the HC condition increases the chain length of C–S–H [37] through hydration of the silicate phases along with the formation of CH [38]. Therefore, the HC condition promoted CH formation as a result of the faster rate of cement hydration, which is consumed when the pozzolanic reaction occurred. However, as shown in Figure 6, the CH consumption was more significant and faster than the CH production, resulting in a lower CH when subjected to a higher temperature. In this study, the TGA method was implemented to measure the CH in all pastes of the corresponding concrete mixes under different curing methods.

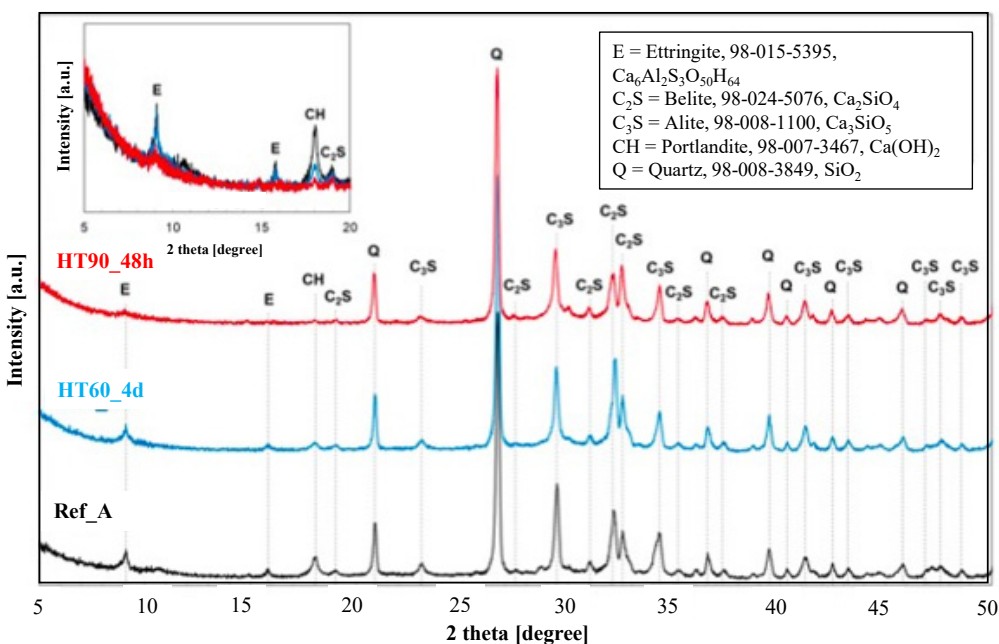

**Figure 6.** XRD results for paste samples tested at 28 days and compared with Ref_A [36].

Figure 7 shows the intensity of the CH peak for mix M3 tested at 28 days. The CH content substantially reduced when cured at a higher temperature. In such a situation, the peak almost disappeared when the curing temperature was increased to 90 °C. At seven days, the CH content in mix M3 under HC conditions was reduced by an average of 70% (the average of three samples; standard deviation was less than 2%). All test results indicated that the increase in the curing temperature caused acceleration in the pozzolanic reactivity. The reduction in CH content of the samples subjected to HC can be consumed in the formation of C–S–H; therefore, concrete with improved mechanical properties was obtained as reported in previous research [39].

As previously mentioned, when the blended concrete (mixes M1, M2, and M3) were heat treated at 90 °C for 8 h, the strength decreased between 7 days and 28 days and further decreased at 90 days. This can be attributed to two factors. The first is the faster pozzolanic activity that occurred during 8 h of heat curing. Thus, most of the water in the samples would have been consumed, which would have densified the microstructure. Therefore, no further pozzolanic reaction would have occurred after completion of heat curing. In addition, when concrete is exposed to a high temperature, the strength is usually reduced, owing to stress relaxation phenomena [28]. When the samples were cured under the other two curing conditions, the strength development continually increased at a much slower rate with time. The highest strength of 162 MPa was obtained at the age of 90 days. The early-age compressive strength of concrete cured either in water curing (WC) or ambient curing (AC) was lower than that in HC. The seven-day compressive strength of WC and AC samples was reduced to 19% and 42%, respectively, compared with the seven-day

strength of the HC samples. Concrete cured under both WC and AC gained strength slowly, and at 28 days, the margin between the two samples was observed in the range of 3–11%. Simultaneously, the AC samples showed 13% lower strength at the age of 28 days compared to the HC samples.

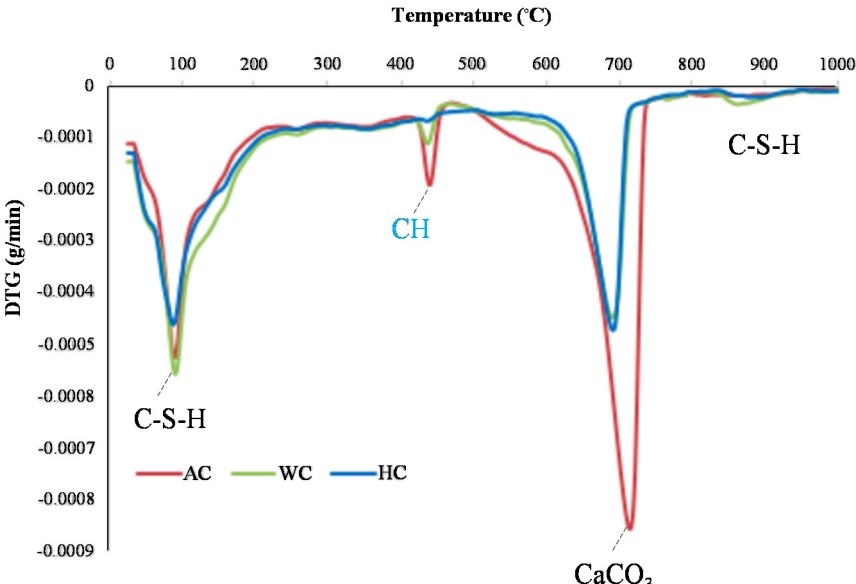

**Figure 7.** TGA results for mix M3 at 28 days.

The 28-day compressive strength of all mixes cured under WC and AC were within the specified compressive strength of above 130 MPa, except for mix M1. AC resulted in compressive strength reductions in most mixes after 28 days and 90 days. However, the reductions were insignificant, at less than 5% at 28 days and 6% at 90 days. The strength ratios of concrete at a particular curing time to the same concrete mix cured for 90 days did not differ significantly between the curing methods. When these ratios were averaged for each curing time and were plotted against the curing time for both curing exposures, as shown in Figure 8, both the 14-day and 28-day strength ratios were higher with AC than with WC, which suggests a higher rate of strength development in AC. Between WC and AC, the compressive strength was insignificant compared with the normal strength. Owing to the excessively dense microstructure and relatively low surface porosity of concrete, water does not penetrate the matrix, which makes external curing ineffective [39]. Several studies have suggested internal curing as an alternative method for improving the compressive strength of concrete [39,40].

It is generalized that the heat curing method reduces the curing time and increases its early strength. However, Yang et al. [41] highlighted that the application of heat curing increases the cost and energy consumption, and restricts the concrete application to the precast concrete industry. Moreover, the impact of WC concrete is minimal, owing to its relatively low surface porosity. Hence, in producing economical and sustainable concrete, the AC method was considered.

### 3.2. Efficiency of SCM Combinations in Concrete

It was observed that the addition of 50% FA content as a substitute for cement reduced the dosage of SP from 4.5% to 1.5% for achieving the target slump of the fresh concrete. However, the high volume of FA content reduced the compressive strength. Therefore, the loss of strength was compensated by adding 10% of either SF or UFCC, which is known as a ternary binder.

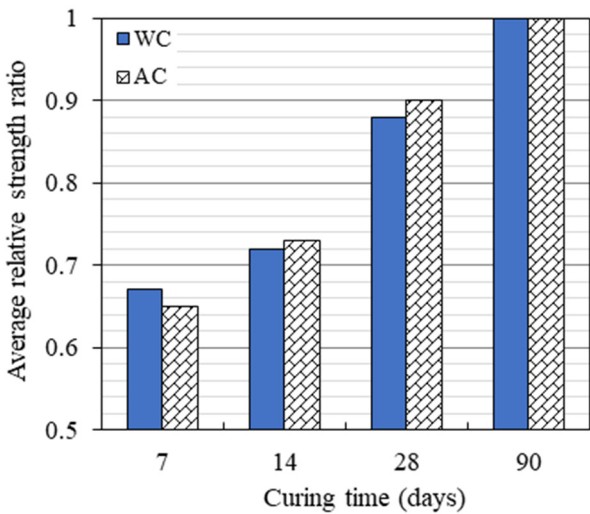

**Figure 8.** Average relative strength ratio versus curing time.

Concrete mixes containing a combination of FA and SF achieved a compressive strength of 55 MPa just after one day, which was estimated to be 34% lower than the strength of mix M0. Various types of SCM, such as FA and SF, have a lower heat of hydration than OPC; therefore, the use of SCM in such concrete reduces the amount of heat evolved, which caused slower strength development at an early age of 24 h. However, at the age of seven days, the compressive strength of mix M2 was approximately 8% lower than that of mix M0 and 27.5% higher than mix M1 with FA and OPC. The compressive strength of mix M1 from 28 days onward exceeded that of mixes M0 and M1. The improvement in compressive strength of mix M2 compared with that of mix M1 was 20% at 28 days and 35% at 90 days. Moreover, the compressive strength graph trend suggested that the high-strength development for mix M2 would continue even after 90 days.

When FA was blended with UFCC, the early strength was improved by 10%. The one-day compressive strength of mix M3 was 27% lower than that of mix M0. However, at the age of 7 days, the compressive strength of mix M3 was almost equivalent to that of mix M0 and 37.6% higher than that in mix M1. These results indicate the effectiveness of UFCC in compensating for low strength at early ages of low-cement concrete. The inclusion of UFCC with a mean particle size of less than 3.5 μm affects the cement hydration in two ways. Finely ground UFCC reacts with the cement and contributes to the formation of the hydration product [42]. In addition, UFCC influences the cement hydration through the effects of fine particles, which act as centers for nucleation [43]. The increased reaction of cement hydration ($C_3S$) and the fine-filler effect of UFCC contributed to the early compressive strength of mix M3, which was greatest at 28 days and 90 days compared with that of the other concrete mixes. The 28-day strength was 135 MPa, which is about 4% higher than that of mixes M0 and M2. The 90-day strength for mix M3 was highest at 160 MPa, which exceeded the strengths of mixes M0, M1, and M2 by 19%, 39%, and 3.2%, respectively.

The synergistic effect between UFCC and high-alumina SCM in a ternary blended system has been reported by Scholer et al. [44]. Arora et al. [45] elaborated that the interaction among various binding materials can maximize the contribution of all constituent materials to the cement performance [36]. Owing to the use of HVFA in producing this low-cement concrete, the inclusion of UFCC was found to be the best combination. An efficiency factor known as the *k* value represents the synergistic efficiency of SCM. When the *k* value is 1, the addition is equivalent to the efficiency of the cement. A *k* value less than 1 indicates lower efficiency of the addition in the compressive strength. The *k* value can be calculated using Equation (1) [46]:

$$k = 1 + \frac{(f_{MA}/f_c) - 1}{p}, \tag{1}$$

where $f_{MA}$ (MPa) indicates the compressive strength of concrete containing the addition, $f_c$ (MPa) indicates the compressive strength of concrete with 100% OPC, and $p$ is content of the addition (%). The $k$ value for all concrete mixes was calculated as shown in Table 6. The highest $k$ value obtained in the ternary blended UFCC system at both 28 and 90 days was 1.08 and 1.39, respectively, followed by those of the ternary blended SF system at 1.0 and 1.36, and the binary blended FA systems at 0.66 and 0.75, respectively. These results indicate that when the HVFA content is incorporated in concrete, SF and UFCC improved the $k$ value by 51–85%. Therefore, the loss in strength from adding FA can be compensated by SF or UFCC to some extent.

**Table 6.** Efficiency factor ($k$ value) for binary or ternary SCM.

| Mix | Binder Combinations | $k$ Value at 28 Days | $k$ Value at 90 Days |
|-----|---------------------|----------------------|----------------------|
| M0 | 100% OPC | - | - |
| M1 | 50% OPC + 50% FA | 0.66 | 0.75 |
| M2 | 50% OPC + 40% FA + 10% SF | 1 | 1.36 |
| M3 | 50% OPC + 40% FA + 10% UFCC | 1.08 | 1.39 |

*3.3. Microstructure Properties of Concrete with Different SCM Combinations*

SEM observation is used to evaluate the microstructure of hardened concrete paste. Information on these microstructures is crucial because it can influence the mechanical performance of the concrete. Figure 9 shows SEM images of the concrete specimens tested at the magnification of 20,000× at 28 days. The microstructure of the control specimen with no replacement (Figure 9a) shows the formation of homogeneous C–S–H gels with neatly arranged small CH crystals.

To develop sustainable concrete, a high amount of SCM is used to minimize its cement content. However, the use of FA to replace a large amount of cement degrades the microstructure of concrete specimens, creating large capillary pores and air voids. Figure 9b shows that the microstructure of concrete is loose when FA is added into the mix, replacing cement by 50%. The dilution effect from the inclusion of FA increased the capillary pores and reduced its compressive strength. The compressive strength for mix M1 was reduced by about 15% compared with that of the reference mix.

The reduction in the strength and microstructure of this binary blend of concrete was improved by combining 10% SF in mix M2 and 10% UFCC in mix M3. Figure 9c shows that the combination of 10% SF in concrete with 40% FA yielded a compact and neat microstructure with minimal visible pores, which contributed to the strength enhancement. Figure 9d shows a much denser microstructure when 10% UFCC was used with 40% FA. The presence of intact FA and UFCC was higher in this mix. This refining process is prominent with the multi-scale particle combination of the FA–UFCC system. The synergistic effect between the UFCC and high-alumina concrete mix significantly improved its efficiency by 60% compared with that in the binary concrete mix, with only FA as a similar replacement. The combination of UFCC in concrete utilizing HVFA was more beneficial than the FA–SF system. The filling effect of UFCC in ultrafine pores in the paste provided additional reinforcement and densified the microstructure of the concrete, which led to more durable concrete.

Figure 10 presents SEM photomicrographs of the concrete samples composed of mixes M0, M1, M2, and M3. These images show the characteristics of the interfacial transition zone in all four types of concrete. Figure 10a represents the 100% OPC concrete mix, showing a wider gap between the aggregate and the paste. The 100% cement mix resulted in a coarse interstitial transition zone (ITZ), which usually affects the long-term performance of concrete. Figure 10b shows the ITZ characteristics of concrete containing 50% OPC and 50% FA. As compared with Figure 10a, the gap between the cement and the aggregate surface was reduced but still had a wider ITZ. Figure 10c,d show concrete containing 40% FA and 10% SF, and 40% FA and 10% UFCC, respectively. The addition of

10% SF or UFCC significantly refined the ITZ. The gap was relatively narrow and difficult to distinguish, particularly in mix M3, which contained 10% UFCC.

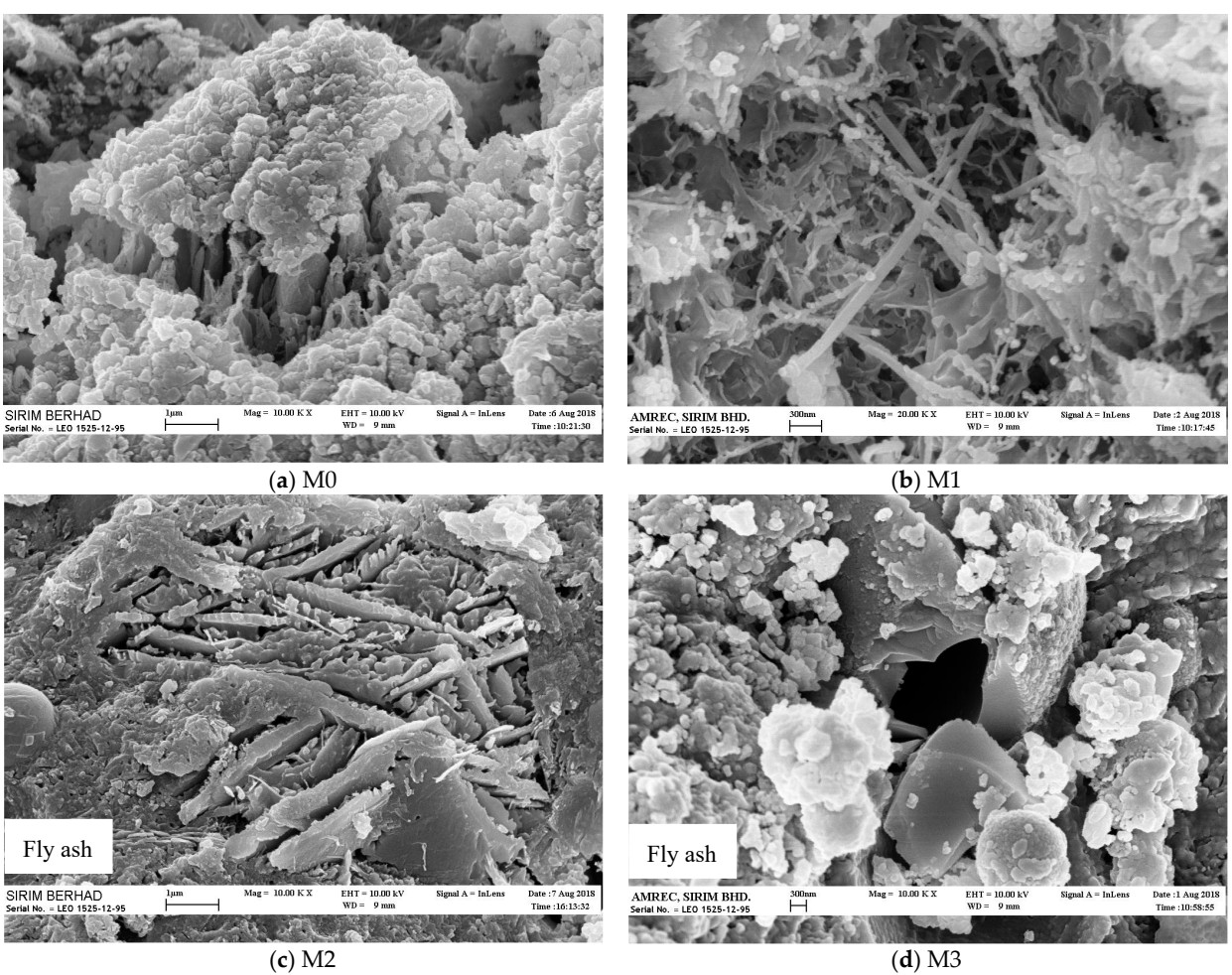

**Figure 9.** Microstructure images of all mixes at 28 days. (**a**) The control mix (**b**) Mix M1 (**c**) fly ash mix M2 (**d**) fly ash mix M3.

Figure 11 shows the XRD patterns of all four mixes at the curing age of 90 days. A comparison of the results in Figures 10 and 11 reveals that the trend shown in Figure 11 validates that observed in Figure 9. It has been established that the strength of cement-based material depends mainly on the strength of the C–S–H gel. However, XRD analysis cannot identify the C–S–H formation in the paste owing to the absence of crystallinity. The highest diffraction of CH is therefore indirectly employed to assess the amount of hydrated products. The XRD analysis tested at 7, 28, and 90 days showed a consistent trend of lower CH peak intensity for mix M3 than for mix M0. The reduction in maximum diffraction of CK was likely caused by the reaction of CH with FA (i.e., pozzolanic reaction) to formulate further hydrated products. In addition, UFCC reacts with aluminate phases in cement, which further reduces the CH value.

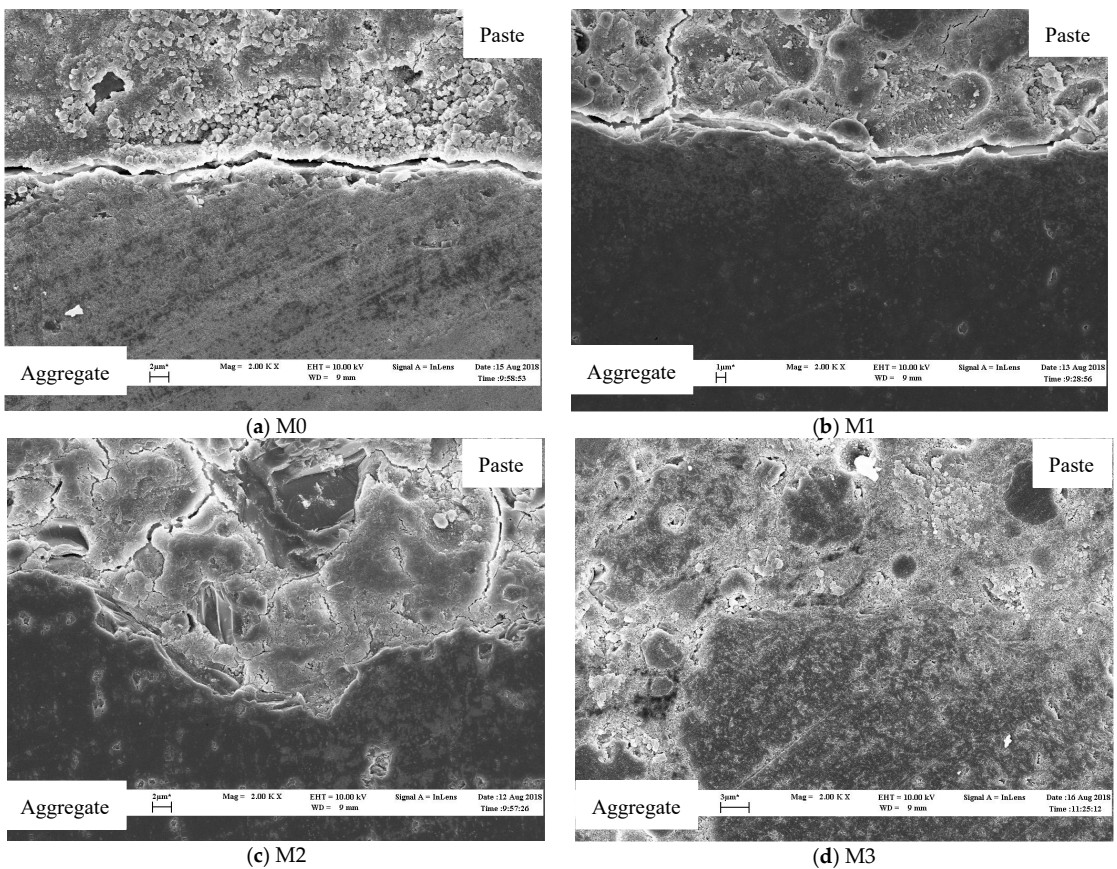

**Figure 10.** Interfacial transition zone (ITZ) of the UHPC at 28 days. (**a**) the control mix (**b**) Mix M1 (**c**) Mix M2 (**d**) Mix M3.

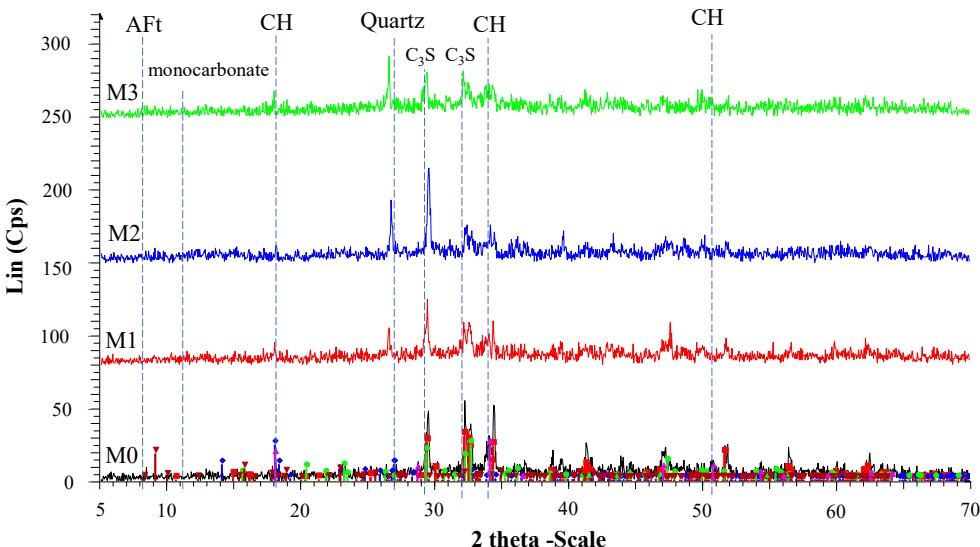

**Figure 11.** XRD pattern of UHPC at 90 days.

### 3.4. Multispectral SEM Analysis

As concluded from the previous results that AC is the optimum curing condition, different paste samples were further analyzed for the control mix (M0), 50% FA (M1), and 40% FA plus 10% SF (M2) at a curing age of 28 days. The results of mix M0 showed that large cement grains remained intact, whereas the smaller ones were surrounded by hydration products, as demonstrated in Figure 12. Similarly, the presence of HVFA in

mix M1 is characterized by various forms and chemical states of FA particles, as shown in Figure 13. The larger particles remained intact, whereas the smaller ones had begun to deteriorate owing to the surface pozzolanic reaction during the curing time, as indicated by the yellow background in Figure 13. The presence of 10% SF and 40% FA in mix M2 as the optimum ternary cementitious binder was investigated, as shown in Figure 14. The role played by SF is clearly demonstrated in the figure as a newly formed type of C–S–H due to the interaction of SF with CH, as obviously indicated in the form of hydration reaction rims around the cement and FA grains. The ternary synergistic interaction among OPC, FA, and SF is evident in this type of analysis.

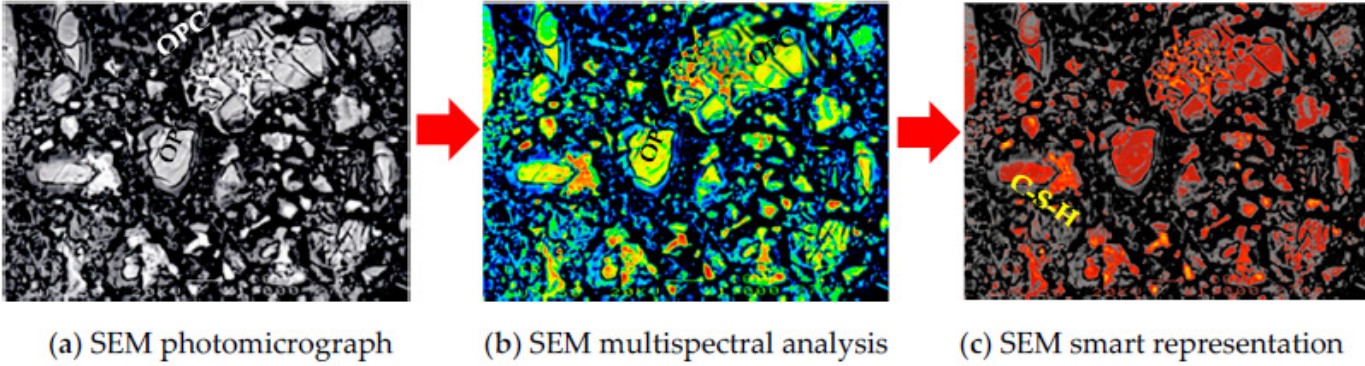

**Figure 12.** SEM multispectral analysis of mix M0.

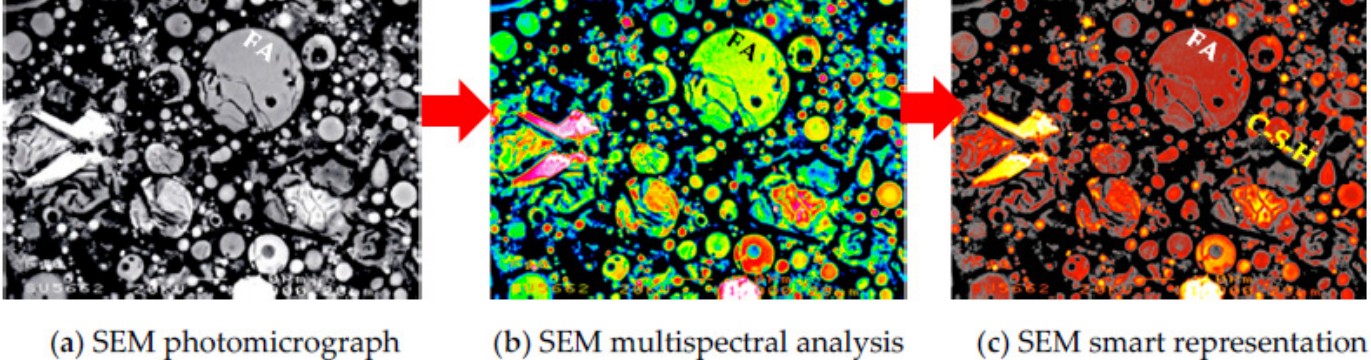

**Figure 13.** SEM multispectral analysis of mix M1.

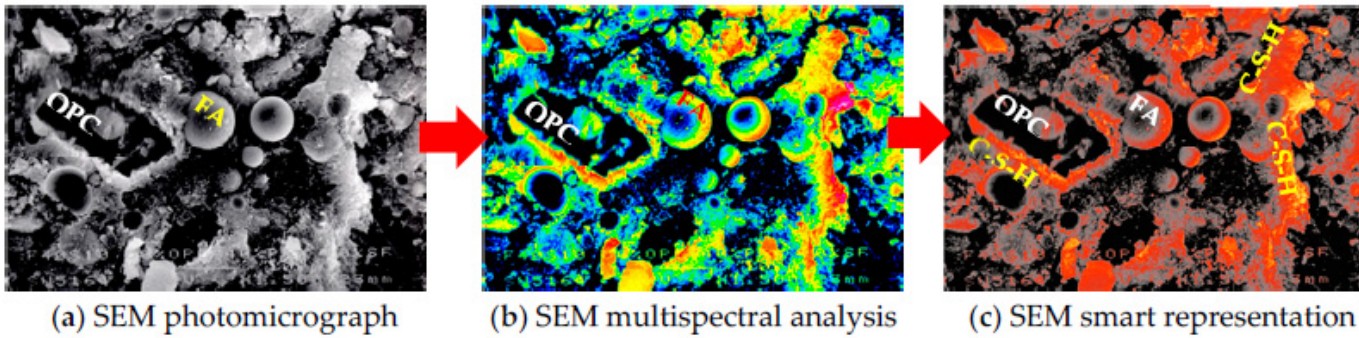

**Figure 14.** SEM multispectral analysis of mix M2.

### 3.5. Tensile and Flexural Strengths

3.5.1. Direct Tensile Strength

Figure 15 shows the average direct tensile strength of all mixes tested at 14 days and 28 days. At 14 days, all mixes containing SCMs provided a direct tensile strength of 3–26% less than that of the reference mix (M0). Mix M1, with 50% FA, resulted in the lowest tensile strength of 4.77 MPa at seven days. However, the direct tensile strength was improved by 17% (5.59 MPa) with 10% SF and by 30% (6.13 MPa) with 10% UFCC including 40% FA. Similarly, at 28 days, mix M1, with 50% FA, showed the lowest strength, whereas mixes M2 and M3 showed about 5% higher tensile strength than that given by the reference mix.

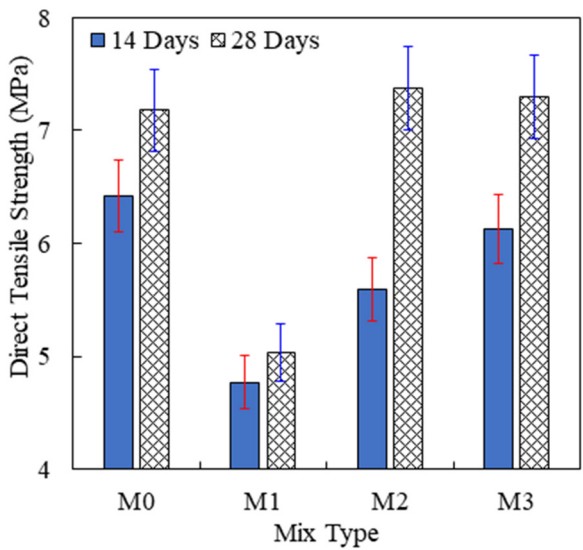

**Figure 15.** Tensile strength of the UHPC mixtures.

The failure pattern for all mixes under direct tensile testing was ductile. No splitting or breakage in the specimens was observed. All specimens were intact at the failure, and the maximum crack width was about 3 mm. This is attributed to the inclusion of 1% steel fibers, which acted as a link to bridge the micro-cracks and prevent crack expansion.

3.5.2. Flexural Strength

Figure 16 shows the typical load-deflection curve for all specimens undergoing the flexural strength test regardless of the mix type. The flexural strength test was performed at 28 days; the results of all mixes are shown in Figure 17. Mix M1, with 50% FA, showed 10% lower flexural strength than the reference mix, M0. In contrast, mix M2, with 10% SF, exhibited 10% higher strength than mix M0. Similarly, mix M3, containing 10% UFCC, showed 18% higher strength than mix M0. When the SF content was 10%, the average 28-day flexural strength of UHPC was 14 MPa; this value is 7% higher than that of mix M0 and about 20% higher than that of mix M1.

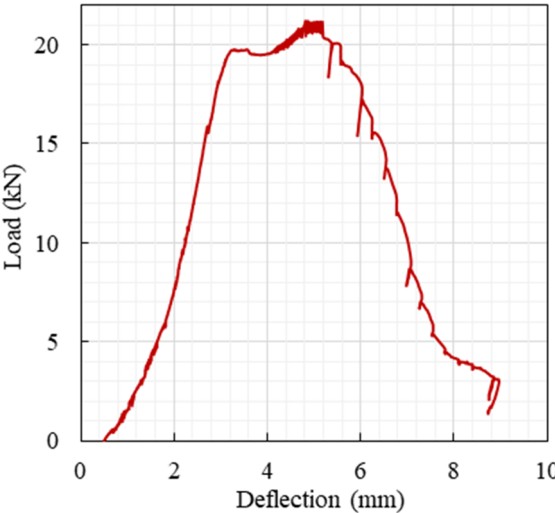

**Figure 16.** Typical load-deflection curve for the developed UHPC.

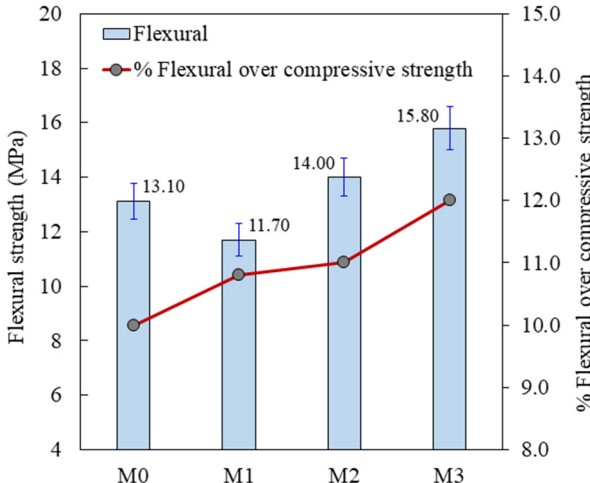

**Figure 17.** Flexural strength of the UHPC mixtures.

The lowest strength with 50% FA is attributed to the latent pozzolanic reactivity of cement, which necessitates a longer cure time. SF and UFCC are composed of much finer particles than those in FA, which resulted in a refined microstructure to provide nucleation centers for a denser microstructure. The finer particles of SF and UFCC filled the pores created by the larger cement particles, which enhanced the OPC–FA interparticle forces and improved the packing density to provide a positive impact on the flexural strength. The tensile and flexural properties of all mixes are associated with the three-dimensional (3D) distribution of the fibers in the matrix. Perfectly oriented (to the stress direction) and homogeneously distributed fibers have a superior likelihood for forming crack bridges, which enhance the entire flexural behavior. Multiple crack patterns were observed in all specimens, as shown in Figure 18. The development of multiple cracks continued up to the peak stress (post-cracking strength), at which point the crack localization occurred. This formation was developed by the effects of fiber inclusion such that the randomly distributed steel fibers controlled the cracks and joined them together.

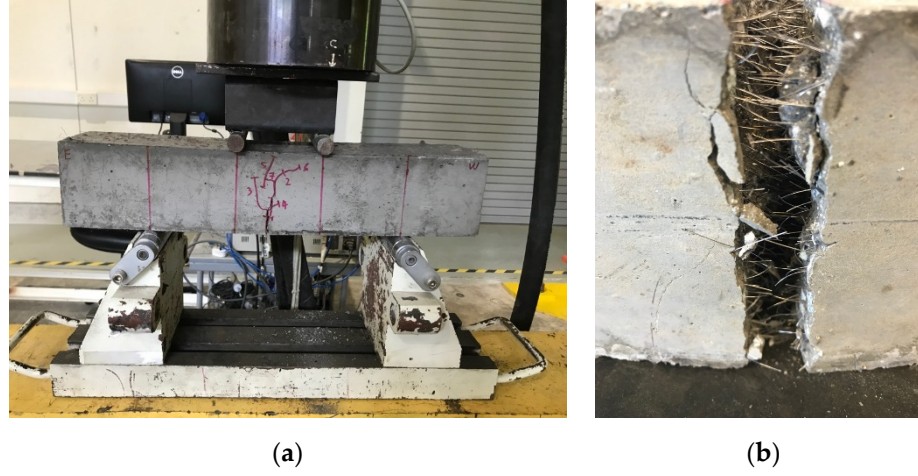

<center>(<b>a</b>)                           (<b>b</b>)</center>

**Figure 18.** Prism after flexural testing: (**a**) flexural testing, and (**b**) cracked surface.

The elasticity limit (i.e., the first crack strength) under tension was obtained from the bending test on concrete prisms; the results are shown in Figure 19. The average limit of elasticity for all mixes was 9 MPa, which indicates a significant strain hardening behavior, as inferred from Figure 16.

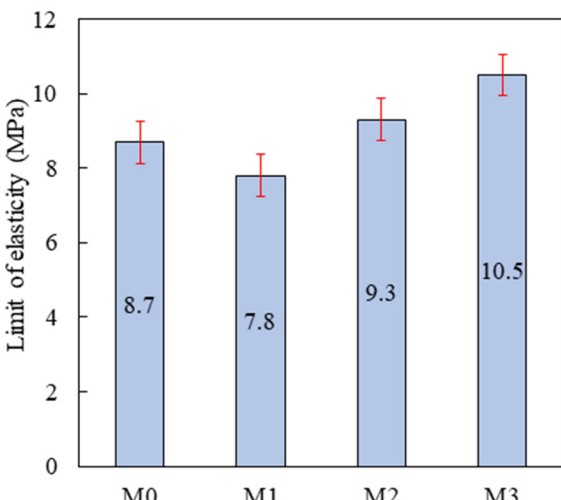

**Figure 19.** Results of the tensile limit of elasticity for the UHPC mixture.

## 4. Conclusions

This study evaluates the effect of UFCC and SF on the compressive strength and microstructure properties of concrete containing HVFA. From the results and discussion, the following conclusions are drawn:

- Up to 50% of cement can be optimally replaced with the combination of FA and UFCC using a W/B of 0.16 without heat treatment.
- The HC treatment significantly improved the seven-day concrete compressive strength by up to 50%. However, the strength was reduced after 28 days owing to weak pozzolanic reaction and relaxation phenomena. Both WC and AC methods showed a slower rate of strength development compared to the HC. AC treatment was useful at 14 and 28 days for the OPC–FA–UFCC concrete system. The concrete achieved the target strength under AC, which eliminated the need for WC and HC.
- HPC can be produced using a high-amount FA replacement and the addition of 10% UFCC with an average particle size of 3.5 μm. The seven-day compressive strength of concrete was improved by 25% with the addition of UFCC in the OPC–FA–UFCC

system, surpassing that of the concrete containing FA only. The synergic effect of efficiency (*k* value) for ternary blend concrete containing OPC–FA–UFCC showed the highest values at all ages, with *k* = 1.06 at 28 days and *k* = 1.39 at 90 days.

- SEM observations showed that the use of UFCC improved the microstructure and ITZ of concrete containing HVFA. The dense matrix was attributed to the possible filling effect, pozzolanic phenomena, and the synergic reaction of FA–UFCC.

**Author Contributions:** Conceptualization, N.S. and M.I.K.; methodology, Y.M.A. and M.O.; investigation, N.A. and G.F.; resources, Y.M.A. and N.S.; data curation, M.O.; writing—original draft preparation, N.A.; writing—review and editing, Y.M.A., N.S. and G.F.; supervision, N.S. and M.I.K. All authors have read and agreed to the published version of the manuscript.

**Funding:** This research and the APC were funded by the Deanship of Scientific Research at King Saud University, grant number RGP-VPP-105.

**Institutional Review Board Statement:** Not applicable.

**Informed Consent Statement:** Not applicable.

**Data Availability Statement:** All research data is available and can be furnished upon request.

**Acknowledgments:** The authors extend their appreciation to the Deanship of Scientific Research at King Saud University for funding the work through the research group project No. RGP-VPP-105. The authors are also pleased to extend acknowledgment to the technical staff of Civil Engineering Departments at King Saud University, Saudi Arabia, and Universiti Teknologi PETRONAS, Malaysia, for their support in conducting the experimental work of this research. The authors thank the Deanship of Scientific Research and RSSU at King Saud University for their technical support.

**Conflicts of Interest:** The authors declare no conflict of interest.

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
