# Peer review of "Enhancing the Microstructure and Sustainability of Ultra-High-Performance Concrete Using Ultrafine Calcium Carbonate and High-Volume Fly Ash under Different Curing Regimes"

_sustainability, doi:10.3390/su13073900_

Round 1

Reviewer 1 Report

The article is well written. The Introduction includes documented literature. I propose some minor additions (see below). The experiment is well documented, the results and discussion are correct and in compliance with the standards of publishing scientific articles.

The discussed issue has been often addressed in the world. Since there are so many articles on this topic, I do not think the article is innovative. However, it contains some novelty effects, so if the authors supplement it with the comments below, I recommend it for publication.

  1. Please, consider whether you wouldn’t like to present works from another part of the world in your Introduction:
  • Materials 2019, 12 (12), 1942, https://doi.org/10.3390/ma12121942
  • Materials 2020, 13 (19), https://doi.org/10.3390/ma13194291
  • Cement and Concrete Research, 124 (2019), 105826, https://doi.org/10.1016/j.cemconres.2019.105826
  1. Figure 7 - What does "Average relative curing ratio" mean? Average from how many samples? How is the average calculated? What is the standard deviation? Average for identical samples? For different ones?
  2. The equation (1) was presented earlier than the publication [44]. The said equation was published by
  • Atis C.D. Cement and Concrete Research, vol. 35, (2005), pp. 1112-1121

Author Response

Reviewer-1

Comment#1: The article is well written. The Introduction includes documented literature. I propose some minor additions (see below). The experiment is well documented, the results and discussion are correct and in compliance with the standards of publishing scientific articles. The discussed issue has been often addressed in the world. Since there are so many articles on this topic, I do not think the article is innovative. However, it contains some novelty effects, so if the authors supplement it with the comments below, I recommend it for publication.

Reply: The constructive criticism, time, and effort to improve the quality of the paper by the esteemed reviewer are highly appreciated. The authors believe that the final version of this work after your fruitful comments and recommendations will meet the standard requirement of your prestigious journal.

Comment#2: Please, consider whether you wouldn’t like to present works from another part of the world in your Introduction:

  • Materials 2019, 12 (12), 1942, https://doi.org/10.3390/ma12121942
  • Materials 2020, 13 (19), https://doi.org/10.3390/ma13194291
  • Cement and Concrete Research, 124 (2019), 105826, https://doi.org/10.1016/j.cemconres.2019.105826.

Reply: Thanks for the recommended valuable references. they have been added in the article’s introduction. Accordingly, the references and bibliography are revised.

Comment#3: Figure 7 - What does "Average relative curing ratio" mean? Average from how many samples? How is the average calculated? What is the standard deviation? Average for identical samples? For different ones?

Reply: We appreciate pointing out this clarification. Kindly, the reported term is the “average relative curing ratio”. Moreover, the standard deviation is reported (the average of three samples and the standard deviation was less than 2%).

Comment#4: The equation (1) was presented earlier than the publication [44]. The said equation was published by:

  • Atis C.D. Cement and Concrete Research, vol. 35, (2005), pp. 1112-1121

Reply: Thanks for the useful comment. Hence, we have cited the indicated reference.

Reviewer 2 Report

Line 188 - "of more than" instead of "at more than"

Line 192 - at the age of 7 days.

Figures 6 and 7 should be replaced by higher resolution graphs.

Line 377 - I think that the authors should refer to Figure 13 instead of 12.

Line 379  - I think that the authors should refer to Figure 13 instead of 14.

Line 380 -  I think that the authors should refer to Figure 14 instead of 12.

Lines 433-435 - what do you understand by "elasticity limit"?

Author Response

Reviewer-2

Comment#1: Line 188 - "of more than" instead of "at more than"

Reply: revised as requested by the respected reviewer.

Comment#2: Line 192 - at the age of 7 days.

Reply: Corrected as commented.

Comment#3: Figures 6 and 7 should be replaced by higher resolution graphs.

Reply: The paper included the best quality of figures (Fig. 6 is currently published elsewhere) we have so far. In the future, we will take this valuable comment into consideration.

Comment#4: Line 377 - I think that the authors should refer to Figure 13 instead of 12.

Reply: Corrected as commented.

Comment#5: Line 379  - I think that the authors should refer to Figure 13 instead of 14.

Reply: Corrected as commented.

Comment#6: Line 380 -  I think that the authors should refer to Figure 14 instead of 12.

Reply: Corrected as commented.

Comment#7: Lines 433-435 - what do you understand by "elasticity limit"?

Reply: Thanks for the valuable remark. It is the first crack strength. This definition is added in the bracket before Figure 19.

Reviewer 3 Report

The paper is interesting and contains many experimental data, which are well interpreted

In the experimental part, steel fibers are used to reinforce the concrete. Corrosion of Steel Fiber Reinforced Concrete in various environment, less harmful as compared to corrosion of steel reinforced concrete, is often considered to be of minor importance, however it exists. It can affect the fibers bridging the cracks and then decrease the strength of the concerned structures.

Did the authors consider the corrosion behavior of the obtained concretes?

In figure 6 it is not clear whether it is the result of the authors' research or it is taken from literature 34. It would also be appropriate to explain what HT90_48h, HT60_4d and Ref_A represent.

Author Response

Reviewer-3

Comment#1: The paper is interesting and contains many experimental data, which are well interpreted

Reply: The time and effort of the esteemed reviewer for improving the quality of this work are deeply appreciated.

Comment#2: In the experimental part, steel fibers are used to reinforce the concrete. Corrosion of Steel Fiber Reinforced Concrete in various environments, less harmful as compared to corrosion of steel reinforced concrete, is often considered to be of minor importance, however it exists. It can affect the fibers bridging the cracks and then decrease the strength of the concerned structures.

Reply: We acknowledge the significance of the point raised by the esteemed reviewer.  In the current study, we have used copper-coated steel fibers to avoid the existence of corrosion. This issue would be investigated in our future research programs.

Comment#3: Did the authors consider the corrosion behavior of the obtained concretes?

Reply: As indicated earlier, the current investigation does not account for corrosion of the fiber, due to their surface coating and the absence of corrosive chemicals such as chloride-bearing chemicals and acids. This issue will be covered in a particular study that covers this issue.

Comment#4: In figure 6 it is not clear whether it is the result of the authors' research or it is taken from literature 34. It would also be appropriate to explain what HT90_48h, HT60_4d and Ref_A represent.

Reply: Thanks to the respected reviewer for the valuable remark. The statement is corrected as follows:

The XRD results of pastes cured for 28 days then heat treated under different temperatures of 20°C (HT20), 60°C (HT60), and 90°C (HT90) (Figure 6) showed reduced peak intensities of both C2S and C3S tested at 28 days compared with the sample cured under normal conditions. Pastes treated under HC at 90°C (HT90) displayed the lowest C2S and C3S values, indicating a higher degree of cement hydration at higher curing temperature. HT treatment promotes cement hydration and pozzolanic reactions significantly and improves the microstructural refinement, as documented in the literature [36].